# A qualitative study of positive psychological experiences and helpful coping behaviours among young people and older adults in the UK during the COVID-19 pandemic

**Liyann Ooi**[ID], **Elise Paul, Alexandra Burton, Daisy Fancourt, Alison R. McKinlay**[ID]*

Research Department of Behavioural Science and Health, Institute of Epidemiology & Health Care, University College London, London, United Kingdom

* a.mckinlay@ucl.ac.uk

## Abstract

While much research has focused on challenges that younger and older people have faced during the COVID-19 pandemic, little attention has been given to the capacity for resilience among these groups. We therefore explored positive psychological experiences and coping behaviours that protected mental health and well-being. Participants were 40 young people (aged 13–24) and 28 older adults (aged 70+) living in the UK during the COVID-19 pandemic. Interviews were held between May 2020 and January 2021. We generated six themes using qualitative thematic analysis, including: engagement in self-fulfilling activities, increased sense of social cohesion, personal growth, use of problem-focused strategies to manage pandemic-related stressors, giving and receiving social and community support, and utilising strategies to regulate thoughts and emotions. While all six themes were relevant both to younger and older adults, there were nuances in how each was experienced and enacted. For example, many older adults adjusted their routines given worries about virus vulnerability, while some young people experienced greater personal growth amidst increased awareness of mental health as they navigated the various lockdown restrictions.

## Introduction

During the novel coronavirus (COVID-19) pandemic, the first UK lockdown was imposed in March 2020 [1], with people being ordered to stay at home unless for essential reasons such as grocery shopping or exercise. People aged over 70 were at first classified as especially vulnerable to the virus [2] and advised to stay at home earlier than other age groups at the start of the pandemic, while young people faced their own unique challenges owing to closures of schools, universities, and workplaces. Evidence suggests that many people in the UK experienced heightened levels of stress, anxiety and fear during the pandemic [3, 4]. However, younger people on average scored highest on measures of loneliness, depression, anxiety, while older adults scored the lowest [5]. Nonetheless, many older adults still experienced complex and adverse psychological responses to events, including depression linked to loneliness [6] and fear arising

---

**Data Availability Statement:** The full dataset cannot be shared publicly because of the ethical risk that this would compromise participant

confidentiality and anonymity. The minimal dataset containing a summary of anonymised participant quotes can be accessed in Supplementary material for this research article.

**Funding:** This research was supported by Nuffield Foundation [WEL/FR-000022583], but the views expressed are those of the authors and not necessarily the Foundation. The study was also supported by the MARCH Mental Health Network funded by the Cross-Disciplinary Mental Health Network Plus initiative supported by UK Research and Innovation [ES/S002588/1], and by the Wellcome Trust [221400/Z/20/Z]. DF was funded by the Wellcome Trust [205407/Z/16/Z]. DF is the Principal investigator, award manager, etc of all funding mentioned in the financial disclosure statement The funders had no role in study design, data collection and analysis, decision to publish, or preparation of the manuscript.

**Competing interests:** The authors declare no competing interests.

from perceived lack of health service availability during the pandemic [7]. Concerns have been raised regarding the disproportionate, long-term psychosocial impacts of COVID-19 on both younger and older adults [8, 9]. Taken together, these factors point to a need for further exploration into the factors that helped buffer the consequences resulting from the pandemic affecting these groups. Such research is important for understanding why some people managed to cope better than others, and for preparing for potential future pandemics.

A narrative review on other infectious disease outbreaks over the last two decades indicated that while adverse psychological consequences were common in previous pandemics, positive psychological experiences were also reported and various coping strategies were used to help deal with the unique challenges of an outbreak [10]. Such positive psychological experiences may include positive emotional, cognitive, social, and behavioural functioning [11]. For example, amid the SARS pandemic, it was reported that some people engaged in healthier lifestyle behaviours and cared more for their own mental health [12, 13]. In the early stages of the COVID-19 pandemic, quantitative findings indicate that people had more opportunities to explore new hobbies, enjoyed more time outdoors and developed greater gratitude for personal relationships [14, 15].

Qualitative research also highlights how people may interpret their pandemic experience positively, including having a greater sense of solidarity with others [16], and a sense of empowerment from increased lifestyle flexibility [17]. The PERMA model of positive psychology [18], which encapsulates positive emotion, engagement, relationships, meaning and achievement, has been used to describe aspects of positive well-being and may be useful when interpreting the positive responses people have reported during the COVID-19 pandemic. However, it has been argued that the model's conceptualisation of well-being does not explicitly address the management of negative emotions pertinent to adverse circumstances of the pandemic [19]. Therefore, recognising positive psychological experiences across domains of the PERMA model in tandem with how individuals deal with negative stressors, may provide a more holistic understanding of well-being of people in younger and older age groups during the pandemic.

How people deal with stressors reflect the concept of coping—crucial psychological and behavioural factors for how individuals draw on behavioural and cognitive efforts to manage stress [20]. Various coping strategies have been theoretically identified [21], including problem-focused coping (i.e., use of planning, restraint, or social support for practical reasons) and emotion-focused coping (i.e., acceptance, denial, positive reinterpretation, or social support for emotions). In the COVID-19 pandemic, the use of socially supported coping (i.e., drawing on social support) has been found to be especially linked with greater improvements in mental health during earlier stages of lockdown restrictions [22], which highlights the key role of social networks in the maintenance of well-being [23]. Moreover, some individual behaviours, such as those involving outdoor activities, have been shown to predict better well-being during the pandemic [24]. Hence the coping behaviours that people engaged in during the pandemic may protect well-being in a way that cuts across and transcends certain coping styles. Furthermore, understanding coping behaviours that individuals perceive as helpful in managing the unique challenges of the pandemic may offer valuable insights into resilience levels during the pandemic.

An evolving body of research suggests that the positive psychological experiences and coping behaviours which protect mental health and well-being during the pandemic may differ across age. In one UK study, young people demonstrated a significantly higher level of positive lifestyle changes than older adults across several domains, such as increased quality time with family, and increased physical exercise [15]. However, when compared to older adults, young people reported greater psychosocial difficulties and more adverse mental health

consequences[25, 26], and some have postulated that part of this is due to significant lifestyle changes owing to the pandemic [27], precarious employment [28] and education interuptions affecting long-term career planning [29]. Nonetheless, it has been observed that young people appeared to show quicker recovery from their symptoms compared to older adults during the initial stages of pandemic restrictions in the UK [30], thereby reflecting the potential for resilience of young people and the need to understand how older adults can be better supported over the longer term. Emerging qualitative findings have lent support to the premise of resilience of older adults and young people respectively as both groups drew on pre-existing and new coping strategies to protect their well-being during the pandemic [31]. Greater sense of coherence (SOC), which encapsulates an individual's ability to comprehend, manage and make sense of a new health threat, has been found to support well-being [32]. While it has been argued that SOC strengthens with age, adverse health impacts in old age may moderate SOC among older adults [33], hence age-related factors may influence capacity to cope with some of life's adversities.

Limited studies to date have explored in depth the experiences of young people and older adults during the COVID-19 pandemic through a positive psychology lens, to investigate the resources that people draw on to support themselves through pandemic-related distress. To address these gaps in the literature, we sought to answer the following question: "what were the positive psychological experiences or coping behaviours that protected mental health and well-being of young people and older adults living in the UK during the COVID-19 pandemic?"

## Methods

### Design

This study forms part of the University College London (UCL) COVID-19 Social Study (CSS), which was the largest panel survey and qualitative interview study of the psychological and social experiences of people in the UK during the COVID-19 pandemic [34]. For the current study, we performed secondary qualitative analysis of 68 interview transcripts reporting the perspectives and experiences of young people (aged 13–24) and older adults (aged 70+). This paper follows the Standards for Reporting Qualitative Research (SRQR) reporting guidelines [35].

### Procedure

Participants were recruited through the CSS e-newsletter, social media, personal contacts, and partner organisations (i.e., third sector services) working with older adults or young people. Eligibility criteria included: aged 70+ or 13–24, living in the UK, and fluent in English. Given the link between certain demographic factors and mental health during the pandemic [30], convenience and purposive sampling strategies were employed to include individuals of diverse age, ethnicity, sex, marital status, living situation and employment status. All participants completed a self-report demographics questionnaire on age, ethnicity, sex, marital status, living situation, employment status, physical and mental health conditions. Semi-structured, one-to-one interviews, lasting between 14 to 85 minutes ($M$ = 45 minutes) were conducted between May 2020 and January 2021, which included the first and second national lockdown in the UK where people experienced cycles of tightened and eased pandemic-related measures (see S1 Table). Interviews were conducted by postgraduate-level, male and female, qualitative health researchers via telephone or video call.

Interviews followed a Topic Guide (see S1 Fig) designed to encompass a range of topics which have been covered in other papers with young people and older adults [7, 29, 36]. The

Topic Guide was developed from supporting theory regarding social networks and SOC [23, 32]. Specific questions were asked to elicit responses on positive experiences during the pandemic and behaviours that helped people cope (see S2 Fig), which formed the focus of this study–although participants also spoke of factors negatively impacting well-being which are reported elsewhere [7, 29]. Participants were offered a £10 shopping voucher as a token of gratitude.

### Research ethics

Ethical approval for the study and research procedure was obtained from the UCL High Risk Ethics Committee (ProjectID:14895/005). We followed best practice guidelines outlined by the Health Research Authority for research involving children and young people when developing our research protocol for obtaining participant consent [37]. We sought written informed consent from participants aged 16 and over. In cases where participants were aged 13–15, they were asked to provide their verbal and written assent and have a parent provide their verbal and written informed consent. All data were held securely and confidentially. Our research protocols were aligned with the principles of the Declaration of Helsinki [38].

### Data analysis

Interviews were audio-recorded, then transcribed verbatim. Transcripts were checked before importing into NVivo 12 [39]. We carried out qualitative thematic analysis with a predominantly deductive and theory-driven orientation during theme and code development [40]. In contrast to *"Big Q"* qualitative research, which tends to be more exploratory and inductive in nature [41], our analysis practices aligned more closely with structured, postpositivist-leaning *"small q"* qualitative research as distinguished by Braun and Clarke (2021) [42]. The lead author (LO) read through all transcripts to ensure completeness of data analysis. While LO coded all transcripts, a second researcher (AM) double-coded three transcripts at the beginning of data analysis and both authors reviewed these identified codes to ensure consistency before completing the remainder of the coding. A deductive coding approach was initially used, whereby a coding framework was first established based on supporting theory on positive psychology and coping [18, 21], and this framework was applied to each transcript through line-by-line coding. The coding framework was then refined iteratively as new concepts were identified by LO. Contradictory remarks and context surrounding codes were noted to draw out subtle nuances. Codes that share a common meaning or concept relevant to the research question were clustered to create themes. Themes and subthemes were developed and regularly discussed between three researchers (LO, EP and AM) throughout the analysis stage to ensure appropriate categorisation of codes. NVivo Crosstab query was used to facilitate analysis of themes across the two age groups.

## Results

Participants were 28 older adults aged 70 to 93 years ($M$ = 77.1, $SD$ = 5.9) and 40 young people aged 13 to 24 years ($M$ = 18.3, $SD$ = 3.4). The majority of participants were White British (75%) and female (57%). Among younger participants, most were in secondary school or university (76%) and living with parent(s) (73%). Among older participants, most were retired (79%) and half were living with a partner/spouse (50%). Forty participants reported having existing physical ($n$ = 31) and/or mental health condition(s) ($n$ = 12). Participant characteristics are summarised in Table 1.

Themes and corresponding sub-themes on positive psychological experiences and helpful coping behaviours are illustrated in Tables 2 and 3 respectively and described below. Many

**Table 1. Self-reported demographic characteristics of participants.**

| Demographic Characteristics | Older Adults ($n = 28$) | Young People ($n = 40$) | Total Participants ($N = 68$) |
|---|---|---|---|
| Age | | | |
| 13–17 | 0 | 21 | 21 |
| 18–24 | 0 | 19 | 19 |
| 70–74 | 11 | 0 | 11 |
| 75–79 | 7 | 0 | 7 |
| 80–84 | 8 | 0 | 8 |
| 85–90 | 1 | 0 | 1 |
| >90 | 1 | 0 | 1 |
| Sex | | | |
| Female | 15 | 24 | 39 |
| Male | 13 | 16 | 29 |
| Ethnicity | | | |
| White British | 23 | 28 | 51 |
| Asian[a] | 3 | 5 | 8 |
| Mixed Race[b] | 0 | 5 | 5 |
| White Other | 2 | 2 | 4 |
| Employment | | | |
| Retired | 22 | 0 | 22 |
| Still at school | 0 | 20 | 20 |
| At university | 0 | 11 | 11 |
| Self employed | 4 | 3 | 7 |
| Full time employment | 0 | 3 | 3 |
| Part time employment | 2 | 0 | 2 |
| Unemployed | 0 | 2 | 2 |
| Apprenticeship | 0 | 1 | 1 |
| Living Situation | | | |
| Live with parent(s) | 0 | 30 | 30 |
| Live with partner/spouse | 14 | 3 | 17 |
| Live alone | 11 | 2 | 13 |
| Live with flat/housemates[c] | 2 | 5 | 7 |
| Live with child(ren) | 1 | 0 | 1 |
| Health Condition | | | |
| Physical health conditon(s)[d] | 20 | 11 | 31 |
| Mental health condition(s)[e] | 4 | 8 | 12 |

*Note.* $n$ = total number

[a]Includes Indian, Pakistani, Other Asian Background

[b]Includes White & Black African, White & Black Caribbean, White & North African, Other Mixed Background

[c]Includes those who live in university halls ($n = 1$) and boarding school ($n = 1$)

[d]E.g., asthma, cancer, diabetes

[e]E.g., anxiety, depression

participants described several positive psychological experiences including engagement in self-fulfilling activities, increased sense of social cohesion and greater personal growth. In addition, many spoke of engaging in various helpful coping behaviours including use of problem-focused strategies to manage pandemic-related stressors, strategies to regulate thoughts and emotions, as well as giving and receiving social and community support.

**Table 2. Themes and subthemes on positive psychological experiences, with summary of similarities and differences between young and older participants.**

| Theme/ Subthemes | Overall | Young People | Older Adults |
|---|---|---|---|
| **Theme 1: Engagement in self-fulfilling activities** | | | |
| Description: Most participants described a 'slower pace of life' during the pandemic which enabled them to engage in activities that brought about feelings of enjoyment or satisfaction | | | |
| **Opportunity for leisure and exploration of new skills** | Both groups said they enjoyed more free time for leisure and to develop new skills. | Young people found renewed sense of competence being able to sharpen their skills. | Older adults found opportunity to rekindle old hobbies and immerse themselves in leisure activities. |
| **Opportunity to organise affairs** | Both groups said they found opportunities to incorporate time for more wellness activities in their lives. | Some young people expressed feelings of satisfaction having the opportunity to spend more time on wellness activities.<br>E.g., "I suppose kind of a sense of satisfaction that it's giving me the space to finally get around to fixing certain aspects of my life. Sorting out routines and getting finally around to doing the exercise. . . meditating, the sort of general wellness personal admin stuff. I'm glad that's happened. . ." (P56, aged 18–24) | Some older adults found opportunity to catch up on overdue tasks in their personal life e.g., sorting out finances and medical care concerning their end of life.<br>E.g., "I've been thinking a lot more about the fact that I'm not going to live that much longer, which I hadn't thought of before, that much. . . .It's something I've done work on before, getting papers in order and writing explanations and things like that, but now that I've got the time to do it, and I'm thinking about it." (P14, aged 75–79) |
| **Theme 2: Increased sense of social cohesion** | | | |
| Description: With the shared experience of going through a pandemic, many participants described a greater sense of togetherness with their community, and more than half of participants reported feeling closer to family and friends as a result of more frequent or consistent contact given a 'greater spirit of loneliness'. | | | |
| **Heightened compassion and connectedness** | Both groups discussed the desire to be more intentional about strengthening or rekindling relationships into the future. | Some young people described increased closeness through consolidated friendship networks.<br>E.g., "it (the pandemic) was like a friends filter" (P27, aged 18–24); "There are some people I haven't necessarily missed not seeing them.. . . I've connected with the people I've needed to. So, in a way my network it's become clearer and seems closer." (P37, aged 18–24) | Some older adults discussed reconnecting with friends or family whom they have grown apart from.<br>E,g., "Yes, those old friends that you'd taken for granted, or didn't seem particularly relevant any more, to realise that if that connection can be rekindled after such a long time, meaningfully, it was worthwhile. So, it's going to be worth making more of in the future, making more effort about in the future. Family, it makes me realise that I do really love them and miss them" (P07, aged 70–74) |
| **Greater sense of community** | Both groups discussed a greater sense of belonging through the "Clap for Carers" (a social movement in appreciation of people working for the UK's National Health Service) and increased interaction with neighbours. | Some young people reported being "more aware of what's going on in my community" (P40, aged 18–24) amidst a greater sense of community.<br>E.g., "I think another positive of the pandemic is that it's brought more people together which sounds cringey but what I mean by that is definitely with our neighbours we've become more close because we're at home all the time. So, a lot of the time if we bake something, like a cake or something, we would give some to our neighbours and then they would do the same for us." (P50, aged 13–17) | Some older adults described how the "Clap for Carers" facilitated increased interaction with neighbours.<br>E.g., "We relate a bit more to the local neighbours. . . . They all come out and clap every Thursday night at 8 o'clock. . . So, chatting more to the neighbours is probably the main positive." (P13, aged 80–84) |
| **Theme 3: Greater personal growth** | | | |
| Description: Some participants recognized self-improvement to their own skills, knowledge, personal qualities and outlook. | | | |
| **New outlook on gratitude** | Both groups described greater appreciation for outdoor activities, use of digital tools to keep in touch with friends and family, and favourable living conditions. | Young people discussed a greater sense of gratitude towards their existing friendships or romantic relationships. | Older adults described feeling fortunate to be retired and financially secure. |

(*Continued*)

**Table 2.** (Continued)

| Theme/ Subthemes | Overall | Young People | Older Adults |
|---|---|---|---|
| **Increased feelings of resilience** | Both groups felt more resilient having navigated unexpected challenges of the pandemic. | Some young people discussed having gained the ability to take better care of themselves. E.g. "I feel like I also increased my ability to look after myself mentally and physically. Like I said with the structure and the regime and things like with being able to make a structure that I can stick to." (P38, aged 18–24) | Some older discussed feeling proud of their capacity to manage their existing physical condition(s) amidst the challenges of the pandemic. E.g., "I'm probably proud of myself, how I've dealt with it (a heart attack)... Sort of managing my own symptoms, managing my own time, and in a way, being strong enough to say, I'm going to go to bed today. Rather than doing the socially acceptable thing of pushing myself... So, I'm quite proud of myself that I've been able to do that." (P65, aged 70–74) |

## Positive psychological experiences

**1 Engagement in self-fulfilling activities.** *1.1 Opportunity for leisure and exploration of new skills*. Many participants described enjoying more free time for leisure and to develop new skills, leading to feelings of improved well-being and satisfaction levels. Among older participants, some found that a slower pace of life during the pandemic allowed them to rekindle old hobbies and immerse themselves in leisure activities: *'I've always loved reading so I've been able to get on with as much reading as I want. . . .to actually just sit for a couple of hours reading and not feel guilty.'* (P06, aged 70–74)

Among younger participants, some expressed a renewed sense of competence being able to sharpen their skills, *'I think I've improved on my music a lot more because I had more time to practice'* (P50, aged 13–17). Some quoted reduced academic pressure or being financially secure as facilitators for being able to learn new skills.

> *I haven't needed to worry about doing a job so I have money next month. So in a way, I've been able to relax a bit which has been really nice and not focused everything on my work. So that is where I've been able to dedicate time to help with anything outside and I actually produce video content for those and learn new skills when it comes to software. . .* (P26, aged 18–24)

*1.2 Opportunity to organise affairs*. Some older adult participants described having the opportunity to catch up on overdue tasks in their personal life including home fixes and *'sort (ing) out paperwork'* for their finances and medical care concerning their end of life.

> *It (the pandemic)'s given me the focus to get into the garden and get on with that because for seven years I was at best maintaining it. Last year I made a big effort to get on top of it. . . .and the lockdown has enabled me to get on with that.* (P06, aged 70–74)

Both young and older people found opportunities to incorporate time for more wellness activities in their life as they began to value their health and well-being more.

> *I suppose kind of a sense of satisfaction that it's giving me the space to finally get around to fixing certain aspects of my life. Sorting out routines and getting finally around to doing the exercise. . . meditating, the sort of general wellness personal admin stuff. I'm glad that's happened. . .* (P56, aged 18–24)

**Table 3. Themes and subthemes on helpful coping behaviours, with summary of similarities and differences between young and older participants.**

| Theme/ Subthemes | Overall | Young People | Older Adults |
|---|---|---|---|
| **Theme 4: Use of problem-focused strategies to manage pandemic-related stressors** Description: Many participants discussed how the pandemic has brought about new stressors in their daily lives and highlighted how coping behaviours targeted at such stressors have been helpful. | | | |
| **Managing intake of pandemic-related news** | Both groups found it helpful to intentionally minimise consumption of news. | Some young people discussed minimising consumption of news in the context of lack of information and certainty about schooling and exams. E.g., "Like with exams and everything, we didn't know anything that was going on and we didn't know when things would be back to normal... Because all you're constantly hearing is, more people have died... this is serious, this is a global pandemic. And it feels... Well, I stopped watching the news... because I just find, it was so intense and depressing..." (P33, aged 13–17) | Some older adults discussed diversifying their source of news to gain different perspectives. E.g., "What I've done is I've taken to listening to the news and things less than before, because I was a bit addicted to the Brexit saga. ...And then in the Brexit period, I listened to it even more. But now, because I know this effect of so much talk about the virus would not be helpful, I just make sure that every evening I've heard the main news. And so, I know what's been said at the government briefing and what other news there is. So, I could listen to world radio. I like to hear the world service because there's a bit more of a perspective." (P17, aged 75–79) |
| **Adopting a new routine** | Both groups discussed having to adapt their lifestyle to 'a socially distant world' and the benefits of a routine. | Young people described how adopting a new routine during the pandemic afforded a sense of structure and helped them feel 'more under control', mitigating the potentially negative impact of remote studying/working from home which blurred study/work-life boundaries. | Older adults described a new sense of purpose and achievement in keeping up a routine as similar as possible to that of pre-pandemic times while allowing themselves the flexibility to 'swap things around' |
| **Theme 5: Use of strategies to regulate thoughts and emotions** Description: Nearly all participants discussed a range of coping strategies to regulate unpleasant thoughts and emotions triggered by circumstances of the pandemic. | | | |
| **Engaging with arts and digital mental health apps** | Both groups discussed how engaging in arts activities (e.g., participating in a "Zoom choir" and listening to music) helped soothe feelings of anxiety or worry and elicited positive emotions. | Young people discussed how digital mental health apps assuaged feelings of worry about uncertainty of their studies and future career. | Older adults missed engaging with arts and cultural activities in-person but most found it helpful being able to continually engage in such activities from home through virtual means during the pandemic. |
| **Being outdoors and connecting with nature** | Both groups reported heeding government guidance that allowed outdoor exercise during lockdown and found it helpful to get 'fresh air' outdoors and connect with nature. | Some young people spoke about how getting out helped them unwind after a long day of working or studying from home. E.g. "I think generally the way that I cope with my stress is going out for walks. ...I spend most of my time indoors, but I think sometimes when you spend too much time inside you just need to go out and breathe fresh air and get a completely different perspective. ...there's just something about staying inside and breathing the same air, I know that you get stuck in your head and you get stuck in your problems. And I feel like when you go outside and you see the outside world, you see that the world is bigger than your problems." (P64, aged 18–24) | Some older adults spoke about how caring for their garden was beneficial to their physical and mental wellbeing. E.g., "And because of Covid I've had to spend more time in the garden and I've certainly found it therapeutic, learning a little more about how you get things to grow. ...And though I'm not becoming a good gardener, because I haven't got fast enough learning skill, I have found being in the open air in the garden very helpful and I think Covid's arrival when it happened, was a huge advantage." (P25, aged 75–79) |
| **Theme 6: Giving and receiving social and community support** Description: Many participants discussed how engaging with social support structures, within their social circles and the wider community, was valuable to good mental health and well-being. | | | |
| **Engaging with interest-based social groups** | Both groups found it rewarding to engage in interest-based social activities. | Young people described how engaging in online multiplayer games strengthened their relationships with friends or expanded their social circle as such games facilitated 'friendly competition' and allowed them to enter into a lot of discussions. | Older adults reported engaging in groups such as walking groups, online writing groups or photography groups to keep their mind and/or body active while maintaining social connections. |

*(Continued)*

**Table 3.** (Continued)

| Theme/ Subthemes | Overall | Young People | Older Adults |
|---|---|---|---|
| **Volunteering and community participation** | Both groups discussed benefits of contributing to the community. | Young people sought out in-person volunteering, such as with foodbanks and delivering essentials to people who are shielding, as 'another excuse to get out of the house' (P57, aged 18–24) while also feeling helpful and needed. | Older adults discussed finding a sense of usefulness while keeping their mind busy by participating in surveys/studies. |

**2 Increased sense of social cohesion.** 2.1 *Heightened compassion and connectedness*. More than half of participants reported feeling closer to family and friends as a result of more frequent or consistent contact, given a *'greater spirit of loneliness'* and heightened sense of compassion for others. Many young and older participants discussed the desire to be more intentional about strengthening or rekindling relationships into the future; *'So, it's going to be worth making more of in the future, making more effort about in the future. Family, it makes me realise that I do really love them and miss them.'* (P07, aged 70–74)

Among young participants, some described increased closeness through consolidated friendship networks as *'it (the pandemic) was like a friends filter'* (P27, aged 18–24); *'There are some people I haven't necessarily missed not seeing them.... I've connected with the people I've needed to. So, in a way my network it's become clearer and seems closer.'* (P37, aged 18–24)

2.2 *Greater sense of community*. With the shared experience of going through a pandemic, many participants described a greater sense of community, especially during the first national lockdown; *'During the initial bits of lockdown, there was quite this sense of solidarity, wasn't it? This general feeling of... this sucks, but we're all in this together. ...That was really nice.'* (P39, aged 18–24)

Several young and older participants discussed a greater sense of belonging through the "Clap for Carers" (a social movement in appreciation of people working for the UK's National Health Service) and increased interactions with neighbours; *'We relate a bit more to the local neighbours.... They all come out and clap every Thursday night at 8 o'clock... So, chatting more to the neighbours is probably the main positive.'* (P13, aged 80–84)

**3 Greater personal growth.** 3.1 *New outlook on gratitude*. Several young and older participants described a new outlook on gratitude, as they had come to *'appreciate all the smaller things'* in view of restrictions that were imposed during the pandemic, including greater appreciation for outdoor activities, use of digital tools to keep in touch with friends and family, and favourable living conditions (i.e., having a garden). Several older participants described feeling fortunate to be retired and financially secure in comparison to many younger, working adults who may be struggling with job security during the pandemic.

*We're retired so we have our pensions, so we're not having to worry about losing our jobs, losing our income or finding a way in which to work in order to be able to earn some money when there can't be any contact between people. So I feel very fortunate that we're not faced with that problem. (P03, aged 70–74)*

Among young participants, many discussed a greater sense of gratitude towards their existing friendships or romantic relationships, given the challenges faced with meeting new people during the pandemic.

*I think one of the downsides of the pandemic was meeting new people wasn't really possible. And one thing I was really grateful for was that I am in a happy relationship with someone for*

*the past two years. But some friends of mine who were single were struggling with this idea of trying to meet new people to date. . . .that made me very grateful that I do have a partner and three different people in my household I can rely on.* (P27, aged 18–24)

3.2 *Increased feelings of resilience.* Having navigated the unexpected challenges of the pandemic, some young and old participants found that they felt more resilient; *'I think I'm stronger than I thought I was, and that I can do things, possibly, I didn't think I would be able to do, and I've coped better than I thought I'd be able to cope.'* (P08, aged 70–74). Some young participants reported having gained the ability to take better care of themselves, thereby feeling more prepared to face future hardships; *'I now know how important it is to have good mental health, constantly. So, that if anything like this were to happen again, I would be prepared.'* (P51, aged 13–17) Some participants with physical or mental health conditions discussed feeling proud of their capacity to manage their condition amidst the challenges of the pandemic.

*I'm probably proud of myself, how I've dealt with it (a heart attack). . . Sort of managing my own symptoms, managing my own time, and in a way, being strong enough to say, I'm going to go to bed today. Rather than doing the socially acceptable thing of pushing myself. . . So, I'm quite proud of myself that I've been able to do that.* (P65, aged 70–74)

## Helpful coping behaviours

**4 Use of problem-focused strategies to manage pandemic-related pressures.**   *4.1 Managing intake of pandemic-related news.* Given the changing circumstances during the pandemic, some participants (both younger and older) acknowledged that *'cutting down on news bulletins but nevertheless, making sure that one is in touch with what is happening, is probably advantageous'* (P04, aged 85–90). Some found it helpful to intentionally minimise consumption of news in view of adverse impacts on their mental health; *'whenever I read the news it's always just bad things. It always makes me worry, so just staying away from that helped as well.'* (P50, aged 13–17)

*4.2 Adopting a new routine.* As the pandemic resulted in restrictions to usual activities, participants reported having to adapt their lifestyle to *'a socially distant world'*. Around one-third of young participants described how adopting a new routine during the pandemic afforded a sense of structure and helped them feel *'more under control'*, mitigating the potentially negative impact of remote studying/working from home which blurred study/work-life boundaries; *'I think the important things to me over lockdown have been actually to give myself a bit of a schedule and wake up by around 7:00, 7:30 in the weekdays than actually have a lie-in.'* (P26, aged 18–24). Furthermore, some discussed how online platforms facilitated self-paced learning which afforded better flexibility and ability to sustain a routine.

Some older adult participants described a new sense of purpose and achievement in keeping up a routine as similar as possible to that of pre-pandemic times while allowing themselves the flexibility to *'swap things around'*.

*I do make myself behave as if, yes, I'm going out to do something. Even if it's only to go for a walk, or do the garden, that is my thing and I'm going to do it properly. . . . and I feel, therefore, positive that I have done things, and achieved things.* (P08, aged 70–74)

**5 Use of strategies to regulate thoughts and emotions.**   *5.1 Engaging with online arts and digital mental health apps.* Many participants, especially older adults, missed engaging with

arts and cultural activities in-person but most found it helpful being able to continually engage in such activities from home through virtual means during the pandemic. Participants reported that engaging in arts activities such as watching performances online or via television, participating in a *"Zoom choir"* and listening to music helped soothe feelings of anxiety or worry and elicited positive emotions.

> *And from Sky Arts on my big TV that has a few programmes a week. . . .compared with how often I might have been going to the theatre or cinema. I've been able to get performances there. It's kept me happy.* (P11, aged 70–74)

Several young people reported worries about uncertainty of their studies and future career; however, digital mental health apps assuaged some of these feelings by helping them to become more aware of strategies to improve their mental health.

> *I now have a mood tracking app (MindDoc) on my phone . . .so that I can track my mood and see what it affects. . . .it can give you very personalised advice. So that's something I've taken up which I think I'm going to keep doing, because I think it's very helpful.* (P61, aged 13–17)

*5.2 Being outdoors and connecting with nature*. The majority of participants reported heeding government guidance that allowed outdoor exercise during lockdown and found it helpful to get *'fresh air'* outdoors, with several describing easy access to green spaces as a facilitator for these coping behaviours. Some young participants spoke about how getting out helped them unwind after a long day of working or studying from home; *'I did go out for the once-a-day exercise. I usually tried to do it after work just to give me that break to change my mindset from work mode to coming back to relaxing.'* (P36, aged 18–24)

While the pandemic seemingly brought everyday life to a standstill, several participants, young and old, found it *'therapeutic'* connecting with nature, where some participants described that seeing plants grow symbolised a sense of time passing.

> *Having plants grow within the house is an amazing way to keep track of time when it all seems so monotonous because it changes, it grows, and you are there. It's a little project and I found that it's so fantastic for me. I've really loved having my plants.* (P26, aged 18–24)

**6 Giving and receiving social and community support.** *6.1 Engaging with interest-based social groups*. Many participants found it rewarding to engage in interest-based social activities. Among these participants, several older adults reported engaging in groups such as walking groups, online writing groups or photography groups to keep their mind and/or body active while maintaining social connections.

> *Oh, and the other one I'm sort of involved, I was before lockdown, in a creative writing group. . . we've been doing that on Zoom.. . . . So that's as a little social thing and keeping the mind active as well. Keep that going, yes.* (P22, aged 75–79)

Amongst young participants, many described how engaging in online multiplayer games strengthened their relationships with friends or expanded their social circle as such games facilitated *'friendly competition'* and allowed them to enter into a lot of discussions; *'Since lockdown started, I've been involved in playing a lot more social board games online. . .and really built a wonderful community of people from all around the world who are really interesting people. . . .It's a nice, different tangent.'* (P26, aged 18–24)

6.2 *Volunteering and community participation.*    Several young participants sought out in-person volunteering, such as with foodbanks and delivering essentials to people who are shielding, as *'another excuse to get out of the house'* (P57, 18–24) while also feeling helpful and needed.

*The only thing that I wasn't doing that was fully rigid with the routine of lockdown was volunteering at the foodbank. And that was because I thought I would just absolutely go crazy if I had to stay in the house the whole time... I just genuinely feel like everything that I do at the moment is really worthwhile to someone, it's always helping someone else in some way...* (P40, aged 18–24)

Among older adult participants, some discussed finding a sense of usefulness while keeping their mind busy by participating in surveys/studies.

*My way of doing my bit... is participating in studies like yours... because I can't go out there and deliver meals to people... but what I can do is contribute to a body of knowledge, that hopefully will be helpful to people in the future.* (P18, aged 70–74)

## Discussion

In this qualitative study, we investigated positive psychological experiences and helpful coping behaviours perceived by young people and older adults in the UK in the first year of the COVID-19 pandemic. There were many similarities in the broad types of psychological experiences that younger and older adults had, as well as similarities in the broad coping behaviours employed, although the details of how these experiences and behaviours were perceived and enacted varied by age. Nonetheless, we identified several common potential components of future interventions to support both young people and older adults in times of pandemics, including activities that foster a sense of community and connectedness (such as "Clap for Carers"), resources that support a stable routine (such as self-paced online learning) and platforms that offer digital means of engaging with hobbies (either on the internet or via mobile phone apps).

One of our salient findings was an increased sense of social cohesion across age groups, congruent with evidence indicating greater gestures of solidarity [16] and the heightened value of personal relationships during the COVID-19 pandemic [14], which reinforce the value of platforms that foster these activities to take place. Our results build on previous research by suggesting how a heightened sense of loneliness and engaging in collective action may encourage a greater community spirit across the age groups amidst social distancing restrictions. While it has been suggested that many young people suffered from loss of social connections during the pandemic [29], a novel finding from our study is that some young participants acknowledged how social restrictions have inadvertently weakened some social ties yet strengthened other key friendships, which facilitated a sense of increased relational closeness. Results suggest that many participants felt that their social networks evolved towards strengthened relationships and stronger community ties, which in turn, improved perceptions of overall well-being.

Several pandemic-related experiences reported by participants appeared to be shaped by their different life stages (i.e., young versus older adulthood) and these factors further shaped subsequent coping behaviours. For instance, younger participants highlighted reduced academic pressure or being financially secure as enablers to exploring new skills. Emerging evidence suggests that younger people, as compared to older adults, had more financial worries

during the pandemic [43] and having financial support has been shown to be a protective factor for coping among young people [44] –hence there is additional need for financial and occupational support toward young people in the future, coming out of the pandemic. The protective role of financial security also resonated with some older participants who described feeling grateful to be retired and financially secure compared with other groups. Moreover, many participants across both age groups discussed a greater sense of appreciation for smaller things in life during the pandemic, reflecting a sense of gratitude which can encourage psychological resilience [45]. Furthermore, our finding that some participants felt a greater sense of resilience within themselves supports the argument that adverse circumstances of living (such as those present during the COVID-19 pandemic) may encourage personal growth and resilience [46]. Our results expand previous findings of personal growth among young people during the pandemic [47] by suggesting that young participants experienced personal growth from having recognised and developed their ability to protect their own mental health in the future.

With regards to helpful coping behaviours, we noted some similarities and differences across age groups. Several young and older participants expressed mental health benefits of engaging in arts and cultural activities remotely, consistent with existing evidence for the use of arts to help cope with emotional experiences during the pandemic [48]. Our results further suggest how increasing digital access to arts during the pandemic is especially valuable to older adults who may otherwise be less likely to engage in arts through digital means as compared to young people. Among young people, our results on the use of digital mental health apps in supporting mental health highlights the promise of such tools in equipping individuals with self-care skills in the future [49]. Evidence suggests that engaging with nature is linked with more positive emotions and fewer symptoms of anxiety, particularly amid stringent lockdown measures during the pandemic [50]. We found that the main differences in this and with several other coping behaviours between young and older age groups were attributed to work-related stress. For instance, some young participants specifically highlighted how spending time outdoors helped them de-stress while adapting to remote working and learning during the pandemic. This could be taken into account by employers and education providers in the event of future social distancing restrictions.

In view of evidence which suggests increased volunteering behaviours in the UK during the COVID-19 pandemic [51], our findings highlight potential age differences in one aspect of volunteering whereby in-person volunteering was especially prominent among young participants. Erikson's stage theory of psychosocial development [52] posits that adolescent years present a conflict of identity versus role confusion whereby adolescents navigate their independence and develop a sense of self. In-person volunteering may have facilitated such aspects of growth as young participants described gaining a sense of being useful while escaping the confines of home during the pandemic. Additionally, young people may have been less worried about their risks from COVID-19 when compared with older adults [53], hence readily sought out in-person volunteering opportunities. Furthermore, a novel finding of our study is that some older adult participants found it beneficial for their wellbeing to volunteer for research surveys during the pandemic, highlighting how volunteering opportunities must be accessible to all age groups. This extends existing literature that being useful in this manner could add meaning to individuals' lives [54], by potentially helping people to make sense of life during the pandemic.

## Strengths and limitations

This is the first known study that qualitatively explored positive psychological experiences and helpful coping behaviours perceived by young people and older adults in the UK during the

COVID-19 pandemic. Purposive sampling enabled greater demographic diversity among participants, which facilitated rich nuances about the experiences of participants that could be targeted in future public health interventions. However, the socioeconomic background of participants was not known, and this may have biased certain findings. While in the present study (where participants were predominantly White British) we found a greater sense of community, some studies indicated reduced sense of social cohesion among some ethnic minority groups in the UK, as the pandemic had more adverse impacts in these communities [55]. Our data collection period may also have influenced the results of our study, given that interviews concluded in January 2021, less than one year since the start of the pandemic as declared by the WHO [56] –hence young and older adults' experiences may be different at later stages of the pandemic. Moreover, the sample may be biased towards people who were less severely impacted by adversities during the pandemic or coping better than others who were unable to participate. Therefore, we cannot assume the transferability of our results to all other contexts and the coping behaviours may not reflect that of the broader population of older or young people.

## Implications

This study highlights some key implications for coping during a pandemic and beyond. First, digital mental health apps may be a valuable tool to support mental health during crisis situations where access to formal mental health support is limited, therefore, further research on their potential is warranted. Second, the benefits of spending time outdoors and connecting with nature may have policy implications for access to green spaces during the pandemic as well as implications for green social prescribing to support well-being [57]. Third, present findings on positive psychological experiences and coping behaviours that protect well-being suggest scope for an extended version of the PERMA model of positive psychology: PERMA-H model [58], which incorporates the facet of positive health. Our findings support the idea that a positive psychology model that includes physical and psychological health may be especially relevant in the context of an infectious disease pandemic to understand well-being more holistically. Moreover, further research is warranted to understand the experience of various socio-economic groups and ethnic backgrounds and capture experiences over time given the fluctuating nature of the pandemic.

## Conclusions

While the COVID-19 pandemic has presented many challenges in various aspects of life of young people and older adults, participants' use of various coping strategies reflect their resilience. In brief, participants described coping behaviours that protect their well-being, with older participants adapting lifestyles in the context of worries about their vulnerability to COVID-19. Moreover, this study highlights positive psychological experiences during the pandemic such as heightened connectedness and increased feelings of resilience. While all six themes were relevant both to younger and older adults, there were nuances in the experience and enactment of behaviours between each age group. These findings may be useful to guide tailored support for well-being among young and older age groups during a pandemic and post-pandemic recovery.

## Supporting information

**S1 Fig. Interview topic guide.**
(DOCX)

**S2 Fig. Interview guide example questions and prompts pertinent to the research question.**
(DOCX)

**S1 Table. Timeline of COVID-19 restrictions in UK between 2020 and 2021 and proportion of study interviews conducted in each month.**
(DOCX)

**S2 Table. Minimal data set.**
(DOCX)

## Acknowledgments

The researchers are grateful for the support of AgeUK, Alzheimer's Society, Healthwise Wales and British Youth Music Theatre during recruitment. We are grateful to Anna Roberts, Joanna Dawes, Louise Baxter, Sara Esser, Rana Conway and Tom May for their help with conducting interviews. The authors extend thanks to study participants for their valuable contributions.

## Author Contributions

**Conceptualization:** Alexandra Burton, Daisy Fancourt, Alison R. McKinlay.

**Formal analysis:** Liyann Ooi.

**Funding acquisition:** Daisy Fancourt.

**Investigation:** Liyann Ooi, Elise Paul.

**Methodology:** Liyann Ooi, Alison R. McKinlay.

**Project administration:** Liyann Ooi.

**Resources:** Alison R. McKinlay.

**Supervision:** Elise Paul, Alison R. McKinlay.

**Writing – original draft:** Liyann Ooi.

**Writing – review & editing:** Elise Paul, Alexandra Burton, Daisy Fancourt, Alison R. McKinlay.

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
