## [Decision Letter · Decision Letter 0]

21 Oct 2022

PONE-D-22-22396A qualitative study of positive psychological experiences and helpful coping behaviours among young people and older adults in the UK during the COVID-19 pandemicPLOS ONE

Dear Dr. McKinlay,

Thank you for submitting your manuscript to PLOS ONE. After careful consideration, we feel that it has merit but does not fully meet PLOS ONE’s publication criteria as it currently stands. Therefore, we invite you to submit a revised version of the manuscript that addresses the points raised during the review process.

I have now received the reports of three Reviewers. All of them find that your manuscript is well written and with important implications in the field. 

It is evident that your work contributes to the in-depth understanding of pandemic experiences for these populations.

Only minor revisions are needed for this paper to be eligible for publication. Please see below.

We look forward to receiving your revised manuscript.

Kind regards,

Eleni Petkari

Academic Editor

PLOS ONE

Journal Requirements:

2. It is in our understanding that the age of emancipation for research in the UK is 18 years of age. In the case of your study we have noted that for participants aged 13-15, they were asked to provide their assent and have a parent provide written informed consent. In your Methods section, please could you provide reference to a legal document indicating that participants above the age of 16 could provide informed consent independently for participation in research, and please could you also indicate whether the IRB approved the informed consent procedure described.

   "This research was supported by Nuffield Foundation [WEL/FR-000022583], but the views expressed are those of the authors and not necessarily the Foundation. The study was also supported by the MARCH Mental Health Network funded by the Cross-Disciplinary Mental Health Network Plus initiative supported by UK Research and Innovation [ES/S002588/1], and by the Wellcome Trust [221400/Z/20/Z]. "

    "This research was supported by Nuffield Foundation [WEL/FR-000022583], but the views expressed are those of the authors and not necessarily the Foundation. The study was also supported by the MARCH Mental Health Network funded by the Cross-Disciplinary Mental Health Network Plus initiative supported by UK Research and Innovation [ES/S002588/1], and by the Wellcome Trust [221400/Z/20/Z]. LO thanks Chevening Scholarships, the UK government’s global scholarship programme, funded by the Foreign, Commonwealth and Development Office (FCDO) and partner organisations University College London and Jeffrey Cheah Foundation for support towards pursuit of a master’s degree that facilitated undertaking of this study. The researchers are grateful for the support of AgeUK, Alzheimer’s Society, Healthwise Wales and British Youth Music Theatre during recruitment. We are grateful to Anna Roberts, Joanna Dawes, Louise Baxter, Sara Esser, Rana Conway and Tom May for their help with conducting interviews. The authors extend thanks to study 

participants for their valuable contributions."

 "This research was supported by Nuffield Foundation [WEL/FR-000022583], but the views expressed are those of the authors and not necessarily the Foundation. The study was also supported by the MARCH Mental Health Network funded by the Cross-Disciplinary Mental Health Network Plus initiative supported by UK Research and Innovation [ES/S002588/1], and by the Wellcome Trust [221400/Z/20/Z]. "

Reviewers' comments:

Reviewer's Responses to Questions

**Comments to the Author**

1. Is the manuscript technically sound, and do the data support the conclusions?

Reviewer #1: Yes

Reviewer #2: Yes

Reviewer #3: Yes

2. Has the statistical analysis been performed appropriately and rigorously? 

Reviewer #1: Yes

Reviewer #2: Yes

Reviewer #3: Yes

3. Have the authors made all data underlying the findings in their manuscript fully available?

Reviewer #1: Yes

Reviewer #2: No

Reviewer #3: Yes

4. Is the manuscript presented in an intelligible fashion and written in standard English?

Reviewer #1: Yes

Reviewer #2: Yes

Reviewer #3: Yes

5. Review Comments to the Author

Reviewer #1: This is an excellent manuscript which coherently details the secondary qualitative analysis of interviews conducted with young and older adults in the UK during the COVID-19 pandemic.

The study aimed to assess the experiences of young and older adults through a 'positive psychology' lens by asking the question “what were the positive psychological experiences or coping behaviours that protected mental health and well-being of young people and older adults living in the UK during the COVID-19 pandemic?”

I believe the qualitative methods that are described in this paper to be methodologically rigorous and well explained. The paper follows the SRQR guidelines for reporting qualitative research.

The results are well presented, including tables which compare and constant younger and older adult experiences.

The discussion provided a in depth review of how the findings of this study can be situated within exisiting literature.

Reviewer #2: I want to thank you for the opportunity to review this interesting study describing the positive psychological experiences and coping behaviors of young people and older adults living in the UK during the COVID-19 pandemic. In my humble opinion, it offers interesting results that the scientific community and society in general can benefit from. I believe that the topic is original, the manuscript is very well written, the analyses seem straightforward, and the results can make a significant contribution to relevant literature. The discussion of the results holds significant suggestions that are highly appreciated.

Reviewer #3: The paper explored positive psychological experiences and coping behaviours that protected mental health and well-being in young people (aged 13-24) and older adults (aged 70+) living

in the UK during the COVID-19 pandemic. The article adequately fits in the profile of Plos One. Nevertheless, I have some minor concerns. In general, it is a useful but a repetitive paper that should be shortened in some parts to facilitate the reading and the comprehension.

1. More information should be given about the interviews, where they took place, how.

2. How exactly the authors assess physical and mental health conditions? They suffer from depression/anxiety during the study/the lockdown or before?

3. All participants were informed about the confidentiality of the information collected? If yes, this sentence should be added

4. Why the second researcher only revise 3 and not all the transcripts in order to be more objective?

5. There is a typo in table 1 footnote: “Carribean” instead of “Caribbean”

6. Clap for Carers” (a social movement in appreciation of people working for the UK’s National Health Service). This explanation should appear in the column “overall”

7. Why Table 1 is divided in several age groups? It is not explained before, and neither it is used in results or discussion section

6. PLOS authors have the option to publish the peer review history of their article (what does this mean?). If published, this will include your full peer review and any attached files.

Reviewer #1: **Yes: **Samuel Cornell

Reviewer #2: No

Reviewer #3: **Yes: **Natalia Martín-María

---

## [Author Response · Author response to Decision Letter 0]

29 Nov 2022

Editor Comments Author Response

1. Please ensure that your manuscript meets PLOS ONE's style requirements, including those for file naming. Thank you - This has been actioned.

2. It is in our understanding that the age of emancipation for research in the UK is 18 years of age. In the case of your study we have noted that for participants aged 13-15, they were asked to provide their assent and have a parent provide written informed consent. In your Methods section, please could you provide reference to a legal document indicating that participants above the age of 16 could provide informed consent independently for participation in research, and please could you also indicate whether the IRB approved the informed consent procedure described.

 We appreciate the editor raising this important point. Adolescents in the UK aged 16-18 with a level of understanding sufficient to agree to participation in research were able to give their informed consent to participate in our research (see UCL ethics guidance here). There is no statute in England, Wales or Northern Ireland governing a child's right to consent to take part in research other than a Clinical Trial of an Investigational Medicinal Product (CTIMP), i.e. consent for non-CTIMPs. We followed these guidelines from the UK Health Research Authority regarding children’s’ involvement with clinical research as best practice.

The UCL Ethics Committee reviewed and approved all of our study procedures and this has been articulated in the Methods section (page 6-7). 

 "This research was supported by Nuffield Foundation [WEL/FR-000022583], but the views expressed are those of the authors and not necessarily the Foundation. The study was also supported by the MARCH Mental Health Network funded by the Cross-Disciplinary Mental Health Network Plus initiative supported by UK Research and Innovation [ES/S002588/1], and by the Wellcome Trust [221400/Z/20/Z]. "

We confirm that this statement is correct and has now been updated and added to our cover letter.

 "This research was supported by Nuffield Foundation [WEL/FR-000022583], but the views expressed are those of the authors and not necessarily the Foundation. The study was also supported by the MARCH Mental Health Network funded by the Cross-Disciplinary Mental Health Network Plus initiative supported by UK Research and Innovation [ES/S002588/1], and by the Wellcome Trust [221400/Z/20/Z]. LO thanks Chevening Scholarships, the UK government’s global scholarship programme, funded by the Foreign, Commonwealth and Development Office (FCDO) and partner organisations University College London and Jeffrey Cheah Foundation for support towards pursuit of a master’s degree that facilitated undertaking of this study. The researchers are grateful for the support of AgeUK, Alzheimer’s Society, Healthwise Wales and British Youth Music Theatre during recruitment. We are grateful to Anna Roberts, Joanna Dawes, Louise Baxter, Sara Esser, Rana Conway and Tom May for their help with conducting interviews. The authors extend thanks to study 

participants for their valuable contributions."

 "This research was supported by Nuffield Foundation [WEL/FR-000022583], but the views expressed are those of the authors and not necessarily the Foundation. The study was also supported by the MARCH Mental Health Network funded by the Cross-Disciplinary Mental Health Network Plus initiative supported by UK Research and Innovation [ES/S002588/1], and by the Wellcome Trust [221400/Z/20/Z]. "

Thank you for pointing this out. Funding related information has now been removed from the acknowledgements section. The correct funding statement has been added to the cover letter. Reference to funding has been removed from the manuscript as requested.

We will update your Data Availability statement to reflect the information you provide in your cover letter. We have now added a minimal dataset as requested with anonymised summaries that helped inform the scientific conclusions drawn on in the manuscript. We are unable to share the full dataset due to the possibility that this could compromise participant anonymity.

Thank you for this suggestion. This information was included in the Methods section; however, we have now included a subsection on Ethics so that this is clearer.

 Many thanks for pointing this out. This has now been added to the end of the manuscript on page 35.

The authors have now re-checked each individual reference source as instructed and confirm that the reference list is correct.

Reviewer Comments: Reviewer 1

Section/Comment Author Response

Overall

This is an excellent manuscript which coherently details the secondary qualitative analysis of interviews conducted with young and older adults in the UK during the COVID-19 pandemic.

The study aimed to assess the experiences of young and older adults through a 'positive psychology' lens by asking the question “what were the positive psychological experiences or coping behaviours that protected mental health and well-being of young people and older adults living in the UK during the COVID-19 pandemic?”

I believe the qualitative methods that are described in this paper to be methodologically rigorous and well explained. The paper follows the SRQR guidelines for reporting qualitative research.

The results are well presented, including tables which compare and constant younger and older adult experiences.

The discussion provided a in depth review of how the findings of this study can be situated within existing literature. 

Many thanks to the reviewer for the positive comments on our manuscript. 

Reviewer 2

Section/Comment Author Response

Overall

I want to thank you for the opportunity to review this interesting study describing the positive psychological experiences and coping behaviours of young people and older adults living in the UK during the COVID-19 pandemic. In my humble opinion, it offers interesting results that the scientific community and society in general can benefit from. I believe that the topic is original, the manuscript is very well written, the analyses seem straightforward, and the results can make a significant contribution to relevant literature. The discussion of the results holds significant suggestions that are highly appreciated. 

Many thanks to the reviewer for the positive comments on our manuscript.

Reviewer 3

Section/Comment Author Response

Overall

The paper explored positive psychological experiences and coping behaviours that protected mental health and well-being in young people (aged 13-24) and older adults (aged 70+) living

in the UK during the COVID-19 pandemic. The article adequately fits in the profile of Plos One. Nevertheless, I have some minor concerns. Many thanks to the reviewer for their helpful suggestions on our manuscript. We have addressed these minor revisions in the attached file and summarise these below.

In general, it is a useful but a repetitive paper that should be shortened in some parts to facilitate the reading and the comprehension. 

Thank you for this feedback. We have shortened some sections that might be seen as repetitive.

Methods

1. More information should be given about the interviews, where they took place, how.

 We thank the reviewer for their suggestions on how we might improve the Methods section of the manuscript. We have re-arranged information about the interviews to make this clearer (page 7). 

2. How exactly the authors assess physical and mental health conditions? They suffer from depression/anxiety during the study/the lockdown or before? 

Physical and mental health data were self-reported, collected via a demographics questionnaire. We have added further detail in the Methods section to make this clearer (page 7). Although some participants spontaneously discussed a new depression or anxiety diagnosis during their interview, we did not routinely ask all participants to specify when their health condition was diagnosed.

3. All participants were informed about the confidentiality of the information collected? If yes, this sentence should be added 

Yes – participants were informed about the confidentiality of information collected. Further text has now been added on page 8.

4. Why the second researcher only revise 3 and not all the transcripts in order to be more objective? 

The dataset is considered extremely large for a qualitative interview study and to have a second researcher review all transcripts would not have been feasible. We did utilize various data triangulation strategies, including attainment of team consensus on themes and inclusion of verbatim quotes, to minimize influence of individual-level researcher biases and thereby enhance trustworthiness of findings (Nowell et al., 2017).

5. There is a typo in table 1 footnote: “Carribean” instead of “Caribbean” 

Thank you very much for pointing out this error, which has now been amended.

6. Clap for Carers” (a social movement in appreciation of people working for the UK’s National Health Service). This explanation should appear in the column “overall” 

 Thank you for this suggestion, which has now been reflected in Table 2 (page 12). 

Results

7. Why Table 1 is divided in several age groups? It is not explained before, and neither it is used in results or discussion section 

Many thanks for raising this. This was done to provide context for interpretation of demographic data eg status of employment and living situation, as well as provide context for interpretation of our findings with regards to different life stages.

---

## [Editor Report · Decision Letter 1]

2 Dec 2022

A qualitative study of positive psychological experiences and helpful coping behaviours among young people and older adults in the UK during the COVID-19 pandemic

PONE-D-22-22396R1

Dear Dr. McKinlay,

We’re pleased to inform you that your manuscript has been judged scientifically suitable for publication and will be formally accepted for publication once it meets all outstanding technical requirements.

Kind regards,

Eleni Petkari

Academic Editor

PLOS ONE
---

## [Editor Report · Acceptance letter]

12 Jan 2023

PONE-D-22-22396R1 

A qualitative study of positive psychological experiences and helpful coping behaviours among young people and older adults in the UK during the COVID-19 pandemic 

Dear Dr. McKinlay:

I'm pleased to inform you that your manuscript has been deemed suitable for publication in PLOS ONE. Congratulations! Your manuscript is now with our production department. 

Kind regards, 

on behalf of

Dr. Eleni Petkari 

Academic Editor

PLOS ONE